# Characterisation of *Metarhizium majus* (Hypocreales: Clavicipitaceae) isolated from the Western Cape Province, South Africa

**Letodi L. Mathulwe**[1], **Karin Jacobs**[2], **Antoinette P. Malan**[1]*, **Klaus Birkhofer**[3], **Matthew F. Addison**[1], **Pia Addison**[1]

**1** Faculty of AgriSciences, Department of Conservation Ecology and Entomology, Stellenbosch University, Matieland, Stellenbosch, South Africa, **2** Faculty of Science, Department of Microbiology, Stellenbosch University, Matieland, Stellenbosch, South Africa, **3** Department of Ecology, Brandenburg University of Technology, Cottbus, Germany

* apm@sun.ac.za

**Data Availability Statement:** All relevant data are within the paper.

## Abstract

Entomopathogenic fungi (EPF) are important soil-dwelling entomopathogens, which can be used as biological control agents against pest insects. EPF are capable of causing lethal epizootics in pest insect populations in agroecosystems. During a survey of the orchard soil at an organic farm, different EPF species were collected and identified to species level, using both morphological and molecular techniques. The EPF were trapped from soil samples taken from an apricot orchard. The traps, which were baited in the laboratory, used susceptible host insects, including the last-instar larvae of *Galleria mellonella* (wax moth larvae) and *Tenebrio molitor* (mealworm larvae). The potential pathogenicity of the local *Metarhizium majus* isolate was tested and verified using susceptible laboratory-reared last-instar *T. molitor* larvae. The identification of the *M. majus* isolated from South African soil was verified using both morphological and molecular techniques. The occurrence of *M. majus* in the South African soil environment had not previously been reported.

## Introduction

Entomopathogenic fungi (EPF), which are cosmopolitan components of the soil microbiota, are commonly isolated from the soil environment for use as biological control agents to manage a broad range of pest insects [1,2]. The genus *Metarhizium* Sorokin (Ascomycetes, Hypocreales) consists of asexually reproducing EPF species, which are characterised by the production of green conidia on the surfaces of infected insect cadavers, and when they are grown on a growth medium [3]. Species belonging to the genus *Metarhizium* are well-studied entomopathogens, which are widely commercialised. Many products derived from the species are on the market for use against a wide range of economically important insect pests of various arthropod orders [4,5]. Such orders include Lepidoptera (leaf miners), Coleoptera (white grubs), Diptera (fruit flies), Orthoptera (locusts and grasshoppers), Hemiptera (whiteflies), Thysanoptera (thrips), and Hymenoptera (ants) [2,4,6,7]. Commercially developed products

**Funding:** KB, PA Ansökan 2015-11 Ekhaga (Ansökan 2015-11) foundation The funders had no role in study design, data collection and analysis, decision to publish, or preparation of the manuscript.

**Competing interests:** The authors have declared that no competing interests exist.

include Real Metarhizium69 (L9281), derived from the *Metarhizium anisopliae* (Metchn.) Sorokin, and Green Muscle (strain IMI 330189, L6198) developed from *M. anisopliae* var. *acridum* (syn. *Metarhizium acridum*) (Driver & Milner) J.F. Bisch., Rehner & Humber [6,7].

Distinguishing between different *Metarhizium* species morphologically is based on their conidial morphology, as using other morphological characteristics is challenging due to the close morphological resemblance involved [3]. *Metarhizium* species are mainly identified and differentiated from each other using molecular techniques [8]. Two main monophyletic groups fall within the *Metarhizium anisopliae* species complex. The PARB clade consists of *Metarhizium pinghaense* Chen & Guo, *Metarhizium anisopliae sensu stricto*, *Metarhizium robertsii* (Metchnikoff) Sorokin and *Metarhizium brunneum* Petch, whereas the MGT clade consists of *Metarhizium majus* Johnst., Bisch., Rehner and Humber and *Metarhizium guizhouense* Chen and Guo [3,9]. The MGT species are distinguished from the PARB clade by means of their relatively large conidia, with *M. majus* having larger cylindrical conidia, relative to *M. guizhouense*, which possess the second largest conidia [9]. *Metarhizium majus* and *M. guizhouense* have been differentiated from each other, based on molecular data, using the translation elongation factor 1 alpha (TEF-1α) gene [3,9].

In the current study, additional information regarding the morphological and molecular evidence obtained is provided to enable the presentation of the first report on the occurrence of *M. majus* in South African soil.

## Materials and methods

### Collection of soil samples and EPF baiting

Soil samples were collected from the orchards surveyed plum, apricot, and quince, at a depth of 15 cm, from under the tree canopy on Tierhoek farm (GPS coordinates: 34˚43'45"S; 19˚47'32"E, in Tierhoek Valley near Robertson in the Western Cape Province. A permit for the collection of soil samples at the farm was issued by the farm owner and manager, B.K.C. Gilson. The samples were obtained at a depth of 15 cm from under the tree canopy. A permit for the collection of soil samples from the farm was issued by the farm owner. The collected soil samples were placed in plastic bags and transferred to a laboratory at Stellenbosch University. Each soil sample was first sifted through a 4-mm mesh sieve to remove the rock and leaf material. After an initial sifting, each soil sample was transferred to a 1-L plastic container, baited with the last-instar larvae of the wax moth *Galleria mellonella* L. (Lepidoptera: Pyralidae) and with *Tenebrio molitor* L. (Coleoptera: Tenebrionidae), namely mealworm, which were kept for 14 days at a room temperature of 25˚C [10–12]. The soil samples were everted after every three days, so as to ensure the penetration of the soil by the insect bait. After every 7 days, the dead insects that showed EPF infection, which was observed in the form of the hardening, or the overt mycosis, of the insect cadaver, were removed from the soil samples. To check the cause of mortality, the dead insects, after having first been washed in sterile distilled water, were then dipped in 75% ethanol for 5 sec, followed by them being dipped twice in distilled water. Each dead insect was placed in a Petri dish fitted with moist filter paper. The Petri dishes were then placed in 2-L plastic containers, fitted with paper towels moistened using sterile distilled water, and incubated at room temperature.

Following a further 7 days of incubation, the spores from the surface of the dead insect cuticles were placed on a Sabouraud dextrose agar plate with 1 g of yeast extract (SDAY), supplemented with 200 μl of Penicillin-Streptomycin, so as to prevent bacterial contamination. After the SDAY plates were sealed and incubated at 25˚C, they were checked for fungal growth for a period of two weeks. The pathogenicity of the fungi cultured on the SDAY for use against insects was verified using the larvae of the wax moth [13].

## Morphological identification

Temporary slides were prepared by means of trapping spores in a drop of water on a glass slide with a coverslip, which was secured with glyceel. The size of the conidia was determined, measuring both the length and the width of 30 spores, using a Zeiss Axiolab 5 light microscope equipped with an Axiocam 208 camera. The scanning electron microscope preparation of spores of different *Metarhizium* species, including *M. majus*, *M. robertsii* (GenBank accession number MT378171), *M. pinghaense* (MT895630), and *M. brunneum* (MT380848), was undertaken and photographed by the Central Analytical Facility of Stellenbosch University. The morphological identification of the entomopathogenic fungi was done according to Humber's key [14].

## Molecular identification

For the purpose of molecular identification, the fungal DNA was extracted from the culture plates using a Zymo research Quick-DNA fungal/bacterial miniprep kit, according to the manufacturer's protocol. A polymerase chain reaction (PCR) was conducted, using the KAPA2G Robust HotStart ReadyMix [KAPA Taq EXtra HotStart DNA Polymerase, KAPA Taq EXtra Buffer, dNTPs (0.3 mM of each dNTP), $MgCl_2$ (2 mM at 1X) and stabilisers] PCR kit. Characterisation was based on the sequencing of the internal transcribed spacer (ITS) region (primers ITS1 and ITS4) and two additional genes, the partial beta-tubulin (BtuB) (primers Bt2a and Bt2b) and the partial TEF-1α (primers EF1F and EF2R) [15,16]. The PCR thermocycle conditions accorded with the technique used by Abaajeh and Nch [17]. The PCR products were visualised on a 1.5% agarose gel in $1 \times$ TBE buffer, using ethidium bromide. A voltage of 92 V for 25 to 30 min was used for the electrophoresis process. The sequences, which were generated by the Central Analytical Facility at Stellenbosch University, were aligned and edited using the CLC main workbench (ver. 8), and BLASTn was carried out on the GenBank database of the National Centre for Biodiversity Information (NCBI) for identification. The fungal cultures (storage number EPF66) were deposited in the fungal collection of the Mycology Unit, Biosystematics Division, Plant Protection Institute, Agricultural Research Council, Pretoria, South Africa.

## Phylogenetic analyses

Phylogenetic analyses were conducted, using the dataset from Rehner and Kepler [16] and Luz et al. [15], concatenate sequences of ITS region, Btub, and TEF-1α genes. The alignments were done employing ClustalX, using the L-INS-I option. The software package Phylogenetic Analysis Using Parsimony (PAUP) [18] was used to construct a neighbour-joining phylogenetic tree, using the uncorrected "p" option. Branch strengths were determined by means of bootstrap analysis (1 000 replicates). A Bayesian analysis was run using MrBayes ver. 3.2.6 [19]. The analysis included four parallel runs of 200 000 generations, with a sampling frequency of 200 generations. The posterior probability values were calculated after the initial 25% of the trees were discarded. The fungal isolates used in the current study to construct the phylogenetic trees are listed in Table 1. The outgroup, *Metarhizium frigidum* (ARSEF 4124[T]), in the construction of the TEF-1α tree were used [16], while for the concatenated generated tree with TEF-1α, ITS and BtuB, *Metarhizium brasilense* (ARSEF 2948[T]) formed the outgroup.

# Results

## Morphological identification

The growth pattern of *M. majus* on the SDAY medium was found to typify the genus *Metarhizium* (Fig 1A). The characteristics of the phialides of *M. majus*, which are cylindrical to

**Table 1. Reference of *Metarhizium* species used in phylogenetic analyses, showing their culture number, isolation source and country of origin, and the GenBank accession numbers of the translation elongation factor 1 alpha (TEF-1α), the beta-tubulin (BtuB) genes and the internal transcribed spacer (ITS) region.**

| Species | Culture number | Isolation source | Country | TEF-1α | BtuB | ITS |
|---|---|---|---|---|---|---|
| *M. acridum* | ARSEF 324 | Orthoptera | Australia | EU248844 | EU248812 | HM055449 |
| *M. acridum* | ARSEF 7486b | Orthoptera | Niger | EU248845 | EU248813 | NR_132019 |
| *M. album* | ARSEF 1942 | Hemiptera | Philippines | KJ398807 | KJ398580 | HM055452 |
| *M. alvesii* | CG1123b | Soil | Brazil | KC520541 | - | - |
| *M. anisopliae* | ARSEF 6347 | Homoptera | Colombia | EU248881 | - | - |
| *M. anisopliae* | ARSEF 7450 | Coleoptera | Australia | EU248852 | EU248823 | HQ331464 |
| *M. anisopliae* | ARSEF 7487b | Orthoptera | Ethiopia | DQ463996 | EU248822 | HQ331446 |
| *M. anisopliae* | CHE CNRCB 235 | Hemiptera | Mexico | KU725694 | - | - |
| *M. anisopliae* | ESALQ1614 | Soil | Brazil | KP027962 | - | - |
| *M. anisopliae* | ESALQ1617 | Soil | Brazil | KP027957 | - | - |
| *M. brasilense* | ARSEF 2948 | Hemiptera | Brazil | KJ398809 | KJ398582 | - |
| *M. brunneum* | ARSEF 2107b | Coleoptera | USA | EU248855 | - | - |
| *M. brunneum* | ARSEF 4179 | Soil | Australia | EU248854 | EU248825 | HQ331451 |
| *M. frigidum* | ARSEF 4124b | Coleoptera | Australia | DQ463978 | EU248828 | NR_132012 |
| *M. guizhouense* | ARSEF 6238 | Lepidoptera | China | EU248857 | EU248830 | HQ331447 |
| *M. guizhouense* | CBS 258.90b | Lepidoptera | China | EU248862 | EU248834 | HQ331448 |
| *M. humberi* | IP 1 | Soil | Brazil | JQ061188 | - | - |
| *M. humberi* | IP 16 | Soil | Brazil | JQ061196 | - | - |
| *M. humberi* | IP 41 | Soil | Brazil | JQ061199 | - | - |
| *M. humberi* | IP 46b | Soil | Brazil | JQ061205 | - | - |
| *M. kalasinense* | BCC53581 | Coleoptera | Thailand | KX823944 | - | - |
| *M. kalasinense* | BCC53582b | Coleoptera | Thailand | KX823945 | - | - |
| *M. lepidiotae* | ARSEF 7412 | Coleoptera | Australia | EU248864 | EU248836 | HQ331455 |
| *M. lepidiotae* | ARSEF 7488b | Coleoptera | Australia | EU248865 | EU248837 | HQ331456 |
| *M. majus* | ARSEF 1914b | Coleoptera | Philippines | KJ398801 | KJ398571 | HQ331445 |
| *M. majus* | ARSEF 1946 | Coleoptera | Philippines | EU248867 | EU248839 | - |
| *M. majus* | TH152 | Soil | South Africa | MT330376 | MT330375 | MT254988 |
| *M. majus* | 2G | Soil | South Africa | MW122513 | - | - |
| *M. pingshaense* | CBS 257.90b | Coleoptera | China | EU248850 | EU248820 | HQ331450 |
| *M. pingshaense* | ARSEF 4342 | Coleoptera | Solomon Islands | EU248851 | EU248821 | HQ331454 |
| *M. robertsii* | ARSEF 23 | Coleoptera | USA | KX342726 | - | - |
| *M. robertsii* | ARSEF 727 | Orthoptera | Brazil | DQ463994 | - | - |
| *M. robertsii* | ARSEF 4739 | Soil | Australia | EU248848 | - | - |
| *M. robertsii* | ARSEF 7501 | Coleoptera | Australia | EU248849 | - | - |
| *M. robertsii* | ESALQ 1621 | Soil | Brazil | KP027980 | - | - |
| *M. robertsii* | ESALQ 1625 | Soil | Brazil | KP027974 | - | - |
| *M. robertsii* | ESALQ 1634 | Soil | Brazil | KP027971 | - | - |
| *M. robertsii* | ESALQ 1635 | Soil | Brazil | KP027977 | - | - |

ellipsoid, can be called *Metarhizium*-like, forming a candelabrum-like arrangement that creates compact conidiophores in a hymenial layer (Fig 1B, 1E and 1F). The conidia of the mature colonies, which were dark green in colour, formed chains of equal length in the clusters obtained (Fig 1B and 1C). The conidia were oblong-elliptical in shape (n = 30), varying 9.0 (7.5–10.2) μm in length and 4.3 (4.0–4.5) μm in width (Fig 1D). The scant difference in the phialides and conidia (except for in terms of size) characterises the *Metarhizium*-like group, with clear differences being found between the *Nomurea*-like and *Paecilomyces*-like groups.

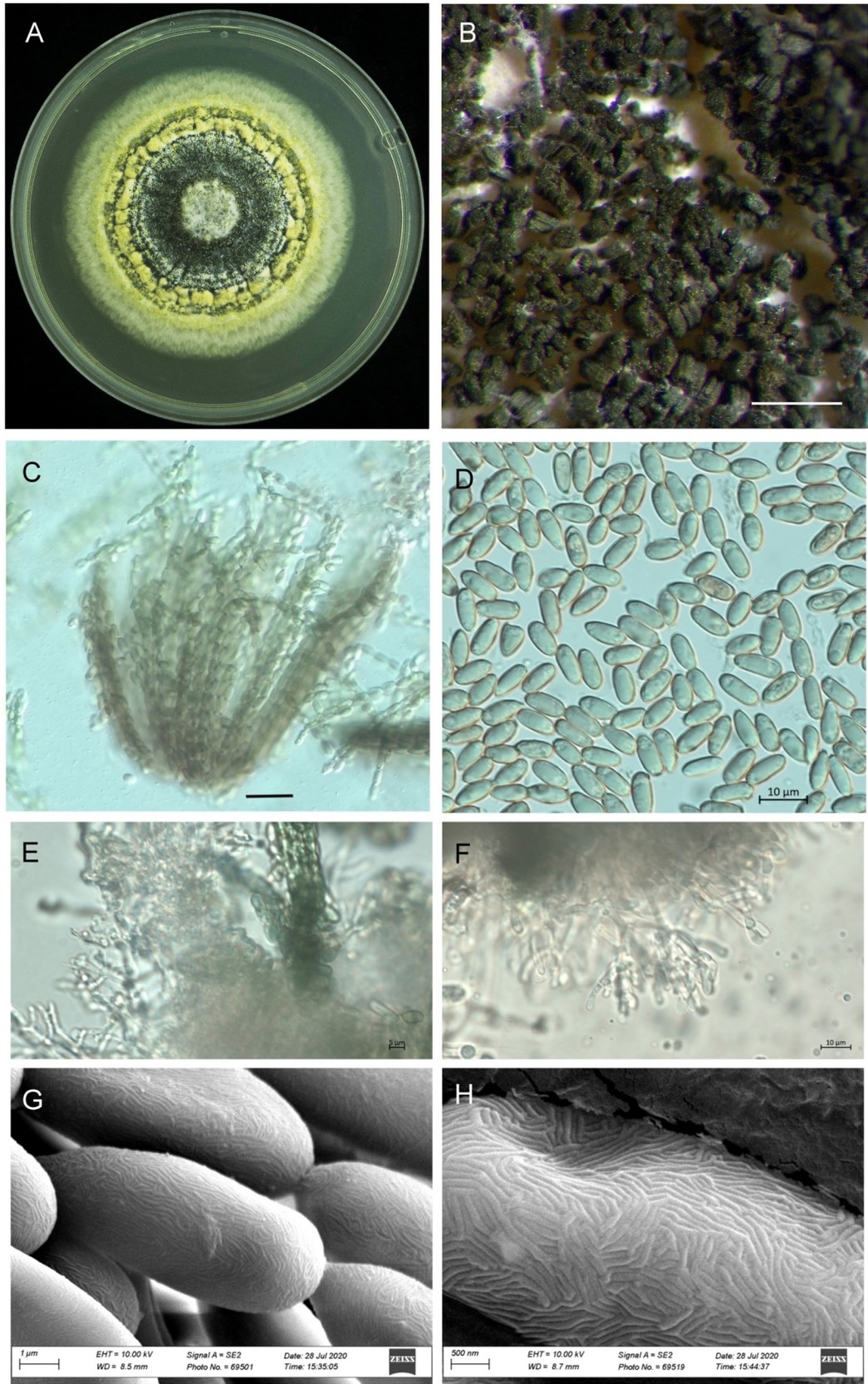

**Fig 1.** Morphology of *Metarhizium majus* TH152, (A) a three-week-old culture on SDAY medium; (B) spores on older plates; (C) bundles of spore strings of the same length; (D) spore shape and size; (E, F) mature phialides with conidiogenous cells and conidia; (G, H) scanning electron microscope picture showing the surface of the conidia. (Scale bars: A = 2 mm; B = 500 μm, C = 5 μm; D-F = 10 μm; G: 10 μm).

The SEM pictures of the four different *Metarhizium* species show no morphological difference in the surface pattern from that of *M. majus* (Fig 1G and 1H).

## Molecular identification

The sequences generated for the *Metarhizium majus* strain collected from an apricot orchard corresponded to those of the type strains. Using the BLASTn function, the ITS region could not differentiate the *M. majus* from *M. anisopliae*. The closest match was found with the type strain of *M. majus* (ARSEF1914/NR152952.1: 98.8%). However, the TEF-1α (with the closest match being *M. majus* ARSEF1914/KJ398801.1: 100%) and the BtuB gene sequences (with the closest match being *M. majus* ARSEF2808/EU248843.1: 99.7%) confirmed the species to be *M. majus*. The sequences obtained were deposited in the GenBank (ITS: MT254988, TEF-1α: MT330376, BtuB: MT330375).

## Phylogenetic analysis

The neighbour-joining phylogeny of the concatenated dataset resulted in a high degree of support for the monophyly of the MGT clades (Fig 2). The *M. guizhouense* was found to form a sister group with *M. majus*, with high percentages of bootstrap support of 87% (Fig 2) and 82% (Fig 3), respectively. The MGT clade formed a sister clade to the PARB clade (Fig 2). The local *M. majus* TH152 isolate, and the two *M. majus* isolates (ARSEF 1914b and ARSEF 1946) collected in the Philippines (Fig 3) grouped in the same clade, with 100% bootstrap confidence. For the TEF-1α gene, *M. majus* showed a 100%, for BtuB a 99.72%, and for ITS a 98.80% identity, with there being, in all cases, 100% coverage, using the BLASTn database of the National Centre for Biotechnology Information (NCBI). The South African *M. majus* did not differ in base similarity from the type strain (KJ398801) of the TEF-1α gene, with it being found to differ by 98% (in terms of 12 base pairs) from the most closely related *M. guizhouense* (EU248862) (Table 2).

## Discussion

The genus *Metarhizium* consists of a diverse group of entomopathogenic fungal species, with a cosmopolitan distribution and a wide range of insect hosts [1,2]. *Metarhizium majus* is considered to be an important potential biological control agent for various insect pests [8]. The fungus is deemed to be an effective biological agent in use against *Odoiporus longicollis* Olivier (Coleoptera: Curculionidae), the banana pseudostem weevil, which is a serious pest affecting banana production [2,20]. The EPF is also used to manage *Oryctes rhinoceros* L. (Coleoptera: Scarabaeidae), the coconut rhinoceros beetle, the activities of which result in major crop losses in coconut and palm oil plantations [21,22].

 The morphological evidence obtained supported the isolate as being *M. majus*, especially in terms of the size of the conidia, which are the largest of all those of the *Metarhizium* species. The growth of the hypha and the phialide morphology is congruent with the genus, with it being difficult to distinguish from the other related species [23]. A previous study indicated that *M. majus* is one of the species in the group with the largest conidia, ranging from 8.5 to 14.5 μl in length and from 2.5 to 3.0 μl in width, with such a characteristic usually being the only usable morphological difference in the group [3,24]. The surface structure of the conidia

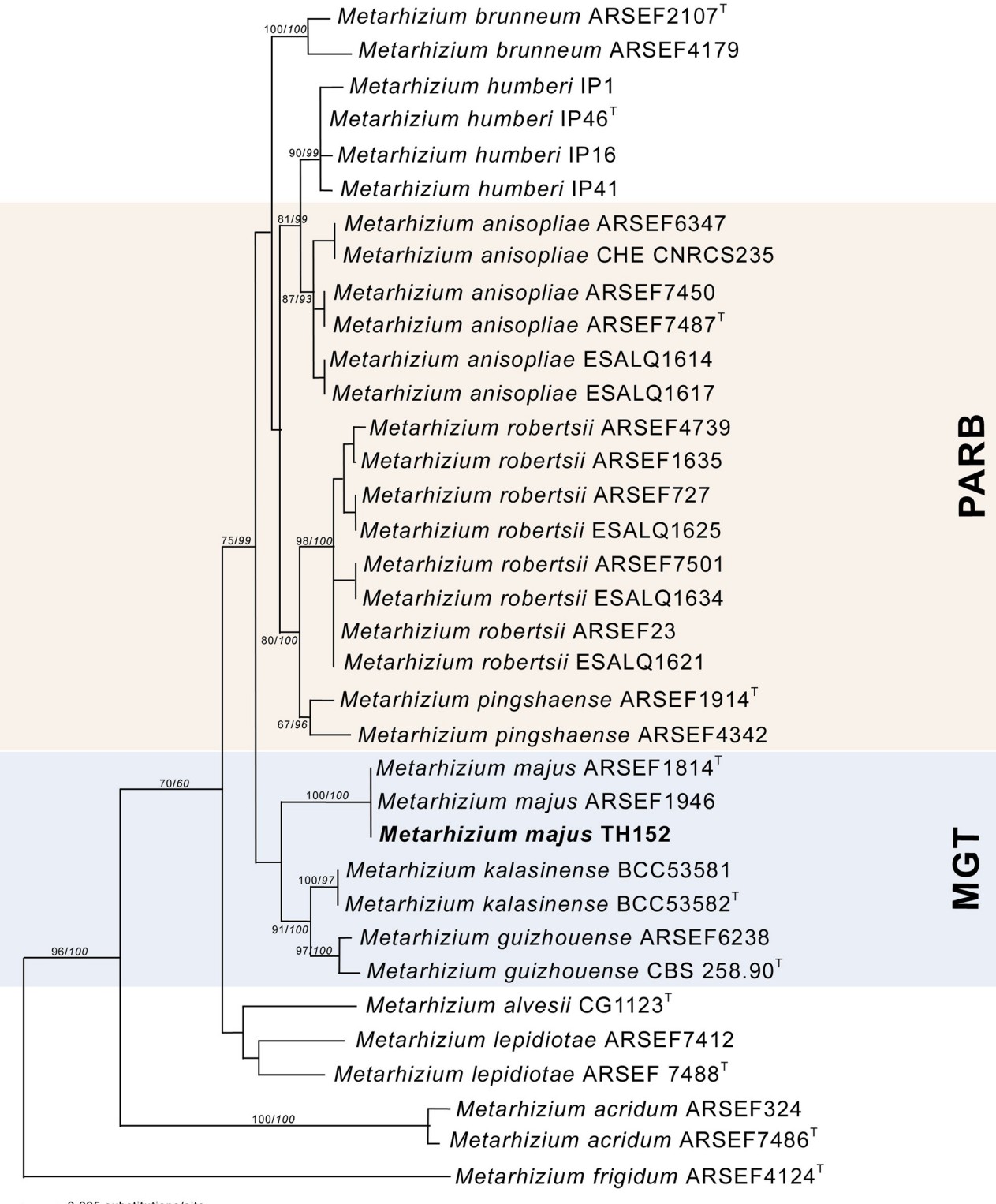

— 0.005 substitutions/site

**Fig 2. The neighbour-joining likelihood phylogenetic tree generated using PAUP with uncorrected "p" option, of *Metarhizium majus* related to the PARB and MGT clades from the analysis of the datasets of 5'intron-rich region of the translation elongation factor 1 alpha (TEF-1α).** Bootstrap values/Bayesian probabilities are denoted above the branch. The tree was rooted, using the sequence from *Metarhizium brasilense* ARSEF2948[T] as outgroup; the isolates with [T] = indicate the type strain.

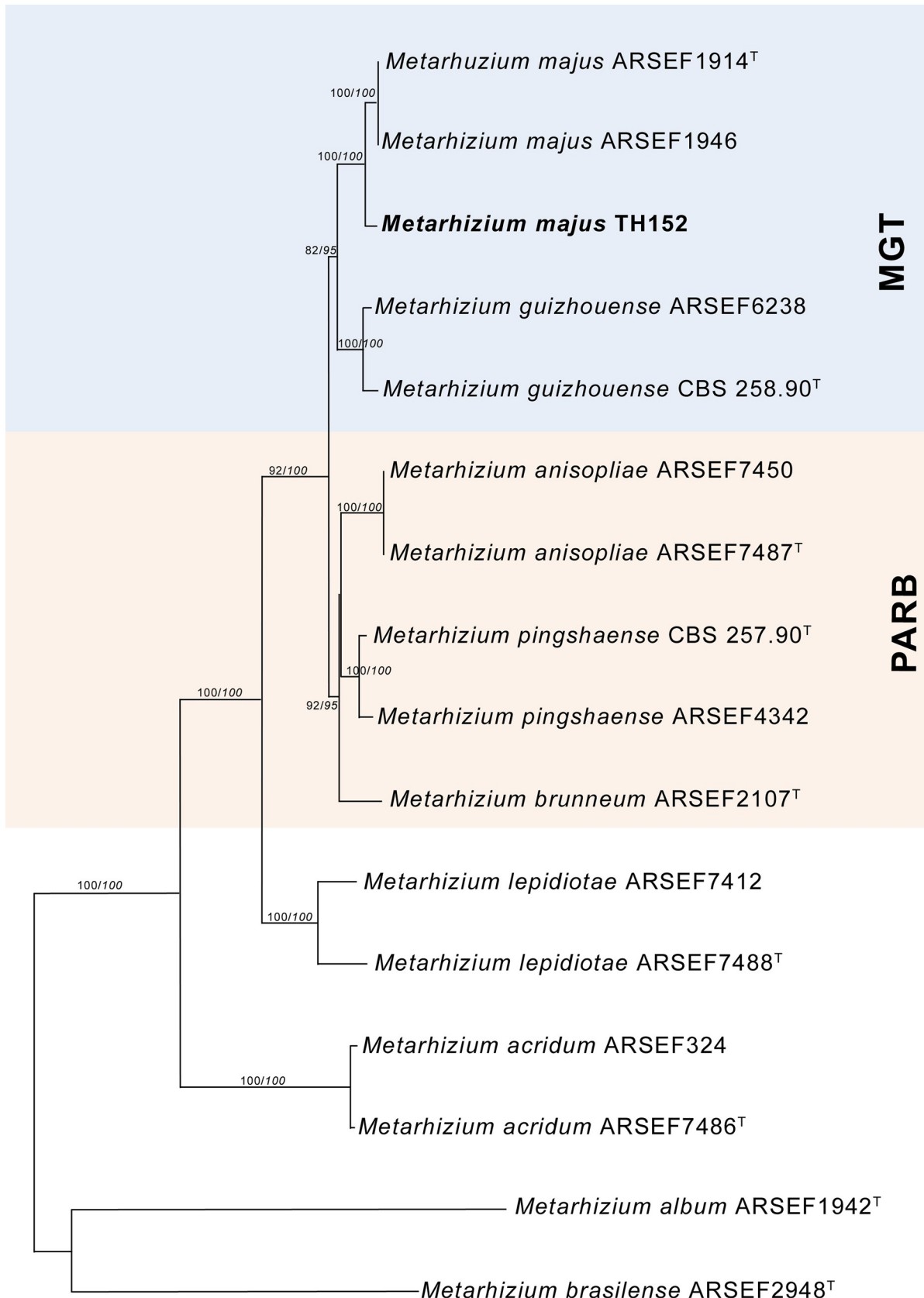

**Fig 3. Neighbour-joining phylogenetic tree generated using PAUP with uncorrected "p" option, of *Metarhizium majus* with regards to related species, based on analysis of the 5'intron-rich region of the translation elongation factor 1 alpha (TEF-1α) gene sequences concatenated with the internal transcribed spacer (ITS) region and the beta-tubulin (BtuB) gene.** The tree was rooted using the sequence from *Metarhizium frigidum* ARSEF 4124[T] as the outgroup. Bootstrap values/Bayesian probabilities are denoted above the branch; isolates with [T] = indicate the type strain.

of *M. majus* was found not to be visually different from *M. robertsii*, *M. pinghaense*, and *M. brunneum*, when subjected to SEM investigation.

The presence of *M. majus* in the soil environment has previously been recorded in other countries, like Japan [9], the USA [2], Australia [25], and Denmark [6]. However, it is the first time that the EPF species concerned has been isolated from South African soil, with the current study providing both morphological and molecular evidence of it being *M. majus*. Unlike other species in the genus, such as *M. anisopliae*, *M. majus* has a narrow to intermediate insect host range [26,27]. Many *Metarhizium* spp. also have the ability to simultaneously colonize roots, which promote plant growth, health, and productivity [27].

The discovery of the South African *M. majus* isolate not only adds new information to the body of knowledge regarding South African soil fungal biodiversity, but opens the way for developing this organism as a product in the local agricultural industry. It has been shown that local strains are generally more effective biocontrol agents, as they are adapted to local environmental conditions, and many regions are tapping into local biodiversity as a source of biopesticides [8,28,29]. The presence of *M. majus* in South Africa, therefore, increases the number of available local EPF isolates that can be used in agricultural ecosystems for the management of insect pests. Its potential as a biocontrol agent, especially Coleoptera [7,20], of which the banded fruit weevil [30] is a key pest in deciduous fruit and grapevine in South Africa, will be investigated in future studies. This is of vital importance in the South African context, as a large proportion of particularly locally produced fruit crops is destined for the European market, which has strict regulations on chemical pesticide use. Developing an arsenal of local biopesticides that can be introduced into a standard integrated pest management program is on par with the global movement towards sustainable agriculture and food safety.

**Table 2. Estimates of evolutionary divergence between type strains of the translation elongation factor 1 alpha (TEF-1α) gene of different *Metarhizium* species.** The number of base pairs difference between the sequences is shown in the form of a matrix, with the standard error above the diagonal. Evolutionary analyses were done in Mega 7.

| Species | | 1 | 2 | 3 | 4 | 5 | 6 | 7 | 8 | 9 | 10 | 11 | 12 | 13 | 14 |
|---|---|---|---|---|---|---|---|---|---|---|---|---|---|---|---|
| 1 | *M. majus* TH152 MT330376 | - | 0.00 | 3.08 | 3.27 | 3.41 | 4.25 | 4.04 | 4.66 | 4.20 | 5.18 | 6.55 | 7.13 | 8.77 | 9.42 |
| 2 | *M. majus* KJ398801 | 0 | - | 3.08 | 3.27 | 3.41 | 4.25 | 4.04 | 4.66 | 4.20 | 5.18 | 6.55 | 7.13 | 8.77 | 9.42 |
| 3 | *M. guizhouense* EU248862 | 12 | 12 | - | 3.48 | 3.46 | 4.39 | 3.90 | 4.79 | 4.02 | 5.61 | 6.95 | 7.55 | 8.69 | 9.44 |
| 4 | *M. anisopliae* DQ463996 | 13 | 13 | 13 | - | 2.11 | 3.76 | 3.23 | 4.67 | 4.79 | 5.04 | 6.22 | 7.27 | 8.73 | 9.24 |
| 5 | *M. pinghaense* EU248850 | 15 | 15 | 13 | 6 | - | 4.01 | 2.84 | 4.70 | 4.57 | 5.35 | 6.44 | 7.29 | 8.56 | 9.59 |
| 6 | *M. lepidiotae* EU248865 | 21 | 21 | 22 | 17 | 19 | - | 4.28 | 4.96 | 5.07 | 5.12 | 6.12 | 7.82 | 9.31 | 9.73 |
| 7 | *M. robertsii* EU248849 | 21 | 21 | 21 | 12 | 9 | 24 | - | 4.85 | 5.13 | 5.54 | 6.64 | 7.29 | 8.45 | 9.36 |
| 8 | *M. humberi* JQ061205 | 23 | 23 | 24 | 17 | 19 | 24 | 25 | - | 3.23 | 4.96 | 6.45 | 6.72 | 8.26 | 8.62 |
| 9 | *M. kalasinense* KX823945 | 23 | 23 | 20 | 24 | 24 | 27 | 32 | 13 | - | 4.96 | 6.30 | 6.77 | 8.16 | 8.77 |
| 10 | *M. alvesii* KC520541 | 25 | 25 | 27 | 24 | 26 | 24 | 32 | 19 | 22 | - | 6.83 | 7.25 | 8.97 | 8.75 |
| 11 | *M. acridum* EU248845T | 48 | 48 | 50 | 47 | 49 | 44 | 53 | 47 | 51 | 51 | - | 8.46 | 9.29 | 9.68 |
| 12 | *M. frigidum* DQ463978 | 61 | 61 | 66 | 59 | 58 | 64 | 57 | 50 | 56 | 60 | 65 | - | 7.22 | 8.24 |
| 13 | *M. album* KJ398807 | 92 | 92 | 97 | 97 | 96 | 98 | 99 | 93 | 93 | 94 | 103 | 93 | - | 8.51 |
| 14 | *M. brasilense* KJ398809 | 97 | 97 | 102 | 96 | 95 | 99 | 92 | 95 | 101 | 96 | 104 | 90 | 92 | - |

## Acknowledgments

The authors are very grateful to the owner of the Tierhoek organic farm. We also wish to thank the staff and students of Stellenbosch University who participated in the soil sampling.

## Author Contributions

**Conceptualization:** Klaus Birkhofer, Matthew F. Addison, Pia Addison.

**Formal analysis:** Letodi L. Mathulwe, Karin Jacobs.

**Funding acquisition:** Klaus Birkhofer, Pia Addison.

**Investigation:** Letodi L. Mathulwe, Karin Jacobs, Antoinette P. Malan.

**Project administration:** Klaus Birkhofer, Pia Addison.

**Supervision:** Pia Addison.

**Writing – original draft:** Letodi L. Mathulwe, Karin Jacobs, Antoinette P. Malan.

**Writing – review & editing:** Letodi L. Mathulwe, Karin Jacobs, Antoinette P. Malan, Klaus Birkhofer, Matthew F. Addison, Pia Addison.

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
