## [Decision Letter · Decision Letter 0]

28 Oct 2020

PONE-D-20-31254

Characterization of Metarhizium majus (Hypocreales: Clavicipitaceae) isolated from the Western Cape province, South Africa

PLOS ONE

Dear Dr. Malan,

Thank you for submitting your manuscript to PLOS ONE. After careful consideration, we feel that it has merit but does not fully meet PLOS ONE’s publication criteria as it currently stands. Therefore, we invite you to submit a revised version of the manuscript that addresses the points raised during the review process.

We look forward to receiving your revised manuscript.

Kind regards,

Ebrahim Shokoohi

Academic Editor

PLOS ONE

Additional Editor Comments:

Dear Dr Malan

I have received the referee comments. The referee comments and the AE comments are provided below. According to the referee comments, your manuscript needs major revisions. However, after point by point addressing all comments of the AE and the referee.

Overall, the paper's morphology, molecular, and phylogeny were provided in a very brief text that needs to be expanded with the most available pieces of information.

1-What model has been used for the phylogeny? Why has NJ been used instead of ML? Do they produce the same result? What is the purpose of Bayesian to be used in this study?

2-In Table 1, if another species sequenced from South Africa needs to be added.

3-Why in Table 1, some locations noted as Brazil and some Brazil, GO? it needs to be clarified. Besides, the outgroup must be added to the table 1.

4-In the result section, the morphological needs to be improved. The authors must compare with any other morphology or morphometrics of the M. majus reported by others. Similar species need to be discussed, as well. Also, the authorship of the species is missed by the authors. Information on the morphology and morphometric must be given in a better format and full.

5-In the phylogenetic, the result needs to be rewritten to indicate the clade. The blast result must be provided in which the similarity should be given.

6-I would suggest adding the Genetic Pairwise distance to compare your sequence and the others for the similarity.

7-In Figure 1 (B and C), the scale is not indicated. By the way, the scale should be the same format for all the figures part.

8-The figure's legends need to be improved with such information as a model, frecA, etc.

9-The quality of Figures 2 and 3 needs to be improved.

10-In Phylogenetic trees, the clades must appear on the figures.

11-Please check all the citations, figures, and the overall format to be in the PLOS ONE style.

Journal Requirements:

Reviewers' comments:

Reviewer's Responses to Questions

**Comments to the Author**

1. Is the manuscript technically sound, and do the data support the conclusions?

Reviewer #1: Partly

2. Has the statistical analysis been performed appropriately and rigorously? 

Reviewer #1: N/A

3. Have the authors made all data underlying the findings in their manuscript fully available?

Reviewer #1: Yes

4. Is the manuscript presented in an intelligible fashion and written in standard English?

Reviewer #1: Yes

5. Review Comments to the Author

Reviewer #1: The authors of the manuscript titled Characterisation of Metarhizium majus (Hypocreales: Clavicipitaceae) isolated from the Western Cape province, South Africa have described the occurrence of an entomopathogenic Metarhizium majus previously unknown to occur in South Africa. The authors carried out pathogenicity assays on two insect pests in addition to morphological and molecular characterization of the fungus.

- Authors should start off by writing in full biological control. This may be shortened later to biocontrol.

- Authors should rewrite the abstract. When rewriting this section, authors should take into consideration the relevant parts of an academic abstract.

- If If Metarhizium spp. are well-studied entomopathogens, why are the authors not providing examples of other commercialized species of the fungus? The authors should also specifically state whether this is a newly identified Metahizium spp., or newly identified as an entomopathogen? Further more, what are the potential benefits of characterizing this new spp. to South Africa and global agriculture?

- Its interesting that throughout the manuscript authors did not consider stating from which orchards soil samples were collected except in line 127. The authors are encouraged to categorically provide the details of the crop and location of the orchards from where soil samples were collected in Western Cape for isolating Metarhizium spp. Were the soil samples bulked, or analysed separately?

- Line 61: Authors should indicate the common name for Tenebrio molitor

- Line 67: Authors should indicate the duration of time for the 75% ethanol and distilled water rinses were carried out.

- How economically important are these insects to South African agriculture? I expect the authors to have focused on characterizing this entomopathogenic fungus using various economically important insect pests. For example, do these two insect pests cut across all the agricultural production regions of South Africa? Are they important constraints to the production of food staples? Do they vector important diseases? These pests are in short not representative enough.

- Line 81: Authors should write in full.

- Line 92: Percent concentration of agarose gel? Voltage used and duration of time for which it was run. What imaging system was used to document the results of the electrophoresis?

- Line 95: BLASTn analysis was carried out on...

- Lines 96-97: Fungal cultures were deposited in the fungal collection of the Biosystems Division of the Agricultural Research Council (ARC), Pretoria.

- What about the partial sequences of the tubulin (BtuB)

- Line 115: What taxonomic key was used for morphological identification?

- Lines 152-153: Authors should provide a citation

_ Line 166: Authors should italicize M. majus

6. PLOS authors have the option to publish the peer review history of their article (what does this mean?). If published, this will include your full peer review and any attached files.

Reviewer #1: No

---

## [Author Response · Author response to Decision Letter 0]

3 Dec 2020

Response to Reviewers

Additional editors comments:

1-What model has been used for the phylogeny? Why has NJ been used instead of ML? Do they produce the same result? What is the purpose of Bayesian to be used in this study?

- The data were analysed using NJ method using uncorrected “p”. Many fungal studies used NJ for phylogenetic comparisons, as it is more robust and can be combined with Bayesian analysis. The purpose of Bayesian analysis is commonly done to test the probability of nodes forming. The ML tree, Bayesian tree and NJ tree had similar topologies in all cases.

2-In Table 1, if another species sequenced from South Africa needs to be added.

- Page 7: Line 128. An additional M. majus isolated, from another locality, have been added to Table 1.

3-Why in Table 1, some locations noted as Brazil and some Brazil, GO? it needs to be clarified. Besides, the outgroup must be added to the table 1.

- Page 7: Line 128. GO next to Brazil deleted. Outgroup added to the table.

4-In the result section, the morphological needs to be improved. The authors must compare with any other morphology or morphometrics of the M. majus reported by others. Information on the morphology and morphometric must be given in a better format and full. Similar species need to be discussed, as well.

- Page 8: Line 132-141. 

- Also, the authorship of the species is missed by the authors. 

- Page 3: Line 57. Authorship is indicated in the introduction section at the first mention of the species name.

5-In the phylogenetic, the result needs to be rewritten to indicate the clade. The blast result must be provided in which the similarity should be given.

- Page 9: Line 157-167. The different clades were described in the text and the Blastn results provide for identify and coverage.

-I would suggest adding the Genetic Pairwise distance to compare your sequence and the others for the similarity.

- Page 10: Line188-188. Table included on genetic pairwise analysis.

7-In Figure 1 (B and C), the scale is not indicated. By the way, the scale should be the same format for all the figures part.

- Corrected – missing scales indicated and made uniform.

8-The figure's legends need to be improved with such information as a model, frecA, etc.

- Legends improved

9-The quality of Figures 2 and 3 needs to be improved.

- Improved quality.

10-In Phylogenetic trees, the clades must appear on the figures.

- Clades indicated in figures

11-Please check all the citations, figures, and the overall format to be in the PLOS ONE style.

- Formatting checked

Reviewers comments

Reviewer #1: 

The authors of the manuscript titled Characterisation of Metarhizium majus (Hypocreales: Clavicipitaceae) isolated from the Western Cape province, South Africa have described the occurrence of an entomopathogenic Metarhizium majus previously unknown to occur in South Africa. The authors carried out pathogenicity assays on two insect pests in addition to morphological and molecular characterization of the fungus.

- Authors should start off by writing in full biological control. This may be shortened later to biocontrol.

Corrected

- Authors should rewrite the abstract. When rewriting this section, authors should take into consideration the relevant parts of an academic abstract.

Page 2: Line 19-30. Abstract adapted

- If If Metarhizium spp. are well-studied entomopathogens, why are the authors not providing examples of other commercialized species of the fungus? 

Page 3: Line 44-50. Information added

The authors should also specifically state whether this is a newly identified Metahizium spp., or newly identified as an entomopathogen? Further more, what are the potential benefits of characterizing this new spp. to South Africa and global agriculture?

Page 11: Line 214-215. Information added. Page11: Line 207-209. Stated that it is a new report for SA, not a new species.

- Its interesting that throughout the manuscript authors did not consider stating from which orchards soil samples were collected except in line 127. The authors are encouraged to categorically provide the details of the crop and location of the orchards from where soil samples were collected in Western Cape for isolating Metarhizium spp. Were the soil samples bulked, or analysed separately

Page 4: Line 69-75. Information added

- Line 61: Authors should indicate the common name for Tenebrio molitor

Page 4: Line 78. Information added

- Line 67: Authors should indicate the duration of time for the 75% ethanol and distilled water rinses were carried out

Page 5: Line 83. Information added

- How economically important are these insects to South African agriculture? I expect the authors to have focused on characterizing this entomopathogenic fungus using various economically important insect pests. For example, do these two insect pests cut across all the agricultural production regions of South Africa? Are they important constraints to the production of food staples? Do they vector important diseases? These pests are in short not representative enough.

Focus are on characterisation and occurrence of M. majus not the insects.

- Line 81: Authors should write in full.

Page 5: Line 97. Corrected

- Line 92: Percent concentration of agarose gel? Voltage used and duration of time for which it was run. What imaging system was used to document the results of the electrophoresis?

Page 6: Line 109-110. Corrected

- Line 95: BLASTn analysis was carried out on...

Page 6: Line 112. Corrected

- Lines 96-97: Fungal cultures were deposited in the fungal collection of the Biosystems Division of the Agricultural Research Council (ARC), Pretoria.

Page 6: Line 113-115. Corrected

- What about the partial sequences of the tubulin (BtuB)

The sequences of Btub were included in the dataset of fig 2. Few sequences available.

- Line 115: What taxonomic key was used for morphological identification?

Page 8: Line 132-142.Information added, however, we relied mainly on molecular identification of the EPF isolate, as the M. anisopliae species complex species mostly look similar morphologically.

- Lines 152-153: Authors should provide a citation

Page 10: Line 193. Corrected

- Line 166: Authors should italicize M. majus

Page 11: Line 206. Corrected

Journal Requirements:

In your Methods section, please provide additional information regarding the permits you obtained for the work. Please ensure you have included the full name of the authority that approved the field site access and, if no permits were required, a brief statement explaining why.

Page : Line 71-73. Information added

---

## [Decision Letter · Decision Letter 1]

4 Jan 2021

PONE-D-20-31254R1

Characterization of Metarhizium majus (Hypocreales: Clavicipitaceae) isolated from the Western Cape province, South Africa

PLOS ONE

Dear Dr. Malan

Thank you for submitting your manuscript to PLOS ONE. After careful consideration, we feel that it has merit but does not fully meet PLOS ONE’s publication criteria as it currently stands. Therefore, we invite you to submit a revised version of the manuscript that addresses the points raised during the review process.

We look forward to receiving your revised manuscript.

Kind regards,

Ebrahim Shokoohi

Academic Editor

PLOS ONE

Additional Editor Comments (if provided):

Dear Authors

I have received the comments from the referee stating that all the issues were not addressed in the revised version of your manuscript. Therefore, I am asking you to check and address all the concerns deeply. The main issue raised by the referee is that the discussion is not written correctly. It needs to be significantly improved. Additionally, the morphology needs to be cited with the proper citation. The language and the manuscript style also needs to be checked carefully. The referee comments attached below.

Kind regards,

Reviewers' comments:

Reviewer's Responses to Questions

**Comments to the Author**

1. If the authors have adequately addressed your comments raised in a previous round of review and you feel that this manuscript is now acceptable for publication, you may indicate that here to bypass the “Comments to the Author” section, enter your conflict of interest statement in the “Confidential to Editor” section, and submit your "Accept" recommendation.

Reviewer #1: (No Response)

2. Is the manuscript technically sound, and do the data support the conclusions?

Reviewer #1: Yes

3. Has the statistical analysis been performed appropriately and rigorously? 

Reviewer #1: Yes

4. Have the authors made all data underlying the findings in their manuscript fully available?

Reviewer #1: Yes

5. Is the manuscript presented in an intelligible fashion and written in standard English?

Reviewer #1: (No Response)

6. Review Comments to the Author

Reviewer #1: Lines 26-27: M. majus and T. molitor should be written in full the first time they are being introduced.

Lines 44-46: Similar to orthoptera, hemiptera and thysanoptera, cite examples for lepidoptera, coleoptera, diptera and hymenoptera.

Line 69: ...from the surveyed plum, apricot, and quince orchards in Tierhoek farm. Soils were obtained at a depth of 15 cm from under the tree canopy.

Line 73: It may not be relevant mentioning the name of the orchard owner at this juncture.

Lines 76-77: wax moth repeated twice

Lines 93-100: what is the taxonomic reference (identification key) that was used for the morphological identification? Although the authors have indicated their reliance on molecular identification, morphological identification is an important aspect of taxonomic classification. If it is an experiment carried out in this study, it carries similar importance as other experiments.

Line 104: The polymerase chain reaction (PCR) was conducted...

Line 110: From what template were the sequences generated? It appears a critical step in the methodology has been ommitted.

Line 118: ...combining 'concatenate' may be a better word to use. In addition, the same synonyms used in the methods section should be used for ITS, BtuB, and EF-1α genes.

Lines 127-128: the same headers listed in line 128 should be used as headers for table 1. In addition include footnotes to table 1 describing in full ITS, BtuB, and EF-1α

Line 166: 'The South African M. majus did not differ in base similarity from the type strain (KJ398801) of the EF-1α gene, and differs by ...% (indicate percentage) from the most closely related M. guizhouense.

Lines 185-189: include footnotes in table 2. For example what does 1 to 14 represent?

The manuscript should be thoroughly checked for grammatical errors. The style of writing and formatting used in other relevant manuscripts an be followed. Articles published in this reputable journal will be suitable.

7. PLOS authors have the option to publish the peer review history of their article (what does this mean?). If published, this will include your full peer review and any attached files.

Reviewer #1: No

---

## [Author Response · Author response to Decision Letter 1]

7 Jan 2021

Reviewer #1: 

Lines 26-27: M. majus and T. molitor should be written in full the first time they are being introduced.

- Page 2: Line28-30. Corrected

Lines 44-46: Similar to orthoptera, hemiptera and thysanoptera, cite examples for lepidoptera, coleoptera, diptera and hymenoptera.

- Page 3: Line 47-49. Corrected

Line 69: ...from the surveyed plum, apricot, and quince orchards in Tierhoek farm. Soils were obtained at a depth of 15 cm from under the tree canopy.

- Page 4: Line 72-73. Changed as suggested

Line 73: It may not be relevant mentioning the name of the orchard owner at this juncture.

- Name of the orchard owner removed.

Lines 76-77: wax moth repeated twice

- Corrected

Lines 93-100: what is the taxonomic reference (identification key) that was used for the morphological identification? Although the authors have indicated their reliance on molecular identification, morphological identification is an important aspect of taxonomic classification. If it is an experiment carried out in this study, it carries similar importance as other experiments.

- Page 5; Line 105-106. Morphological of identification of the entomopathogenic fungi was done by according to the key Humber 2012. Add to reference list: 14. Humber RA. Identification of entomopathogenic fungi. In: Lacey LA, editor. Manual of techniques in invertebrate pathology. 2nd ed. Burlington: Elsevier; 2012. pp. 151-187. https://doi.org/10.1016/B978-0-12-386899-2.00006-3

Line 104: The polymerase chain reaction (PCR) was conducted...

- Page 5: Line 110. Corrected

Line 110: From what template were the sequences generated? It appears a critical step in the methodology has been ommitted.

- Page 5-6: Line 110-112. A polymerase chain reaction (PCR) was conducted, using the KAPA2G Robust HotStart ReadyMix [KAPA Taq EXtra HotStart DNA Polymerase, KAPA Taq EXtra Buffer, dNTPs (0.3 mM of each dNTP), MgCl2 (2 mM at 1X) and stabilisers] PCR kit.

Line 118: ...combining 'concatenate' may be a better word to use. In addition, the same synonyms used in the methods section should be used for ITS, BtuB, and EF-1α genes.

- Page 6: Line 127. Corrected as suggested

Lines 127-128: the same headers listed in line 128 should be used as headers for table 1. In addition include footnotes to table 1 describing in full ITS, BtuB, and EF-1α

- Page 7: Line 138-139. Corrected throughout text

Line 166: 'The South African M. majus did not differ in base similarity from the type strain (KJ398801) of the EF-1α gene, and differs by ...% (indicate percentage) from the most closely related M. guizhouense.

- Page 9: Line178-181. Corrected as suggested

Lines 185-189: include footnotes in table 2. For example what does 1 to 14 represent?

- The cell-column with the numbers have been merged with those of the species name to indicate that the number correspond with the horizontal heading of the matrix, which show the standard error in the above diagonal.

The manuscript should be thoroughly checked for grammatical errors. The style of writing and formatting used in other relevant manuscripts an be followed. Articles published in this reputable journal will be suitable.

- Language edited by a professional language editor, style thoroughly checked and the discussion improved

---

## [Editor Report · Decision Letter 2]

12 Jan 2021

PONE-D-20-31254R2

Characterization of Metarhizium majus (Hypocreales: Clavicipitaceae) isolated from the Western Cape province, South Africa

PLOS ONE

Dear Dr. Malan,

Thank you for submitting your manuscript to PLOS ONE. After careful consideration, we feel that it has merit but does not fully meet PLOS ONE’s publication criteria as it currently stands. Therefore, we invite you to submit a revised version of the manuscript that addresses the points raised during the review process.

We look forward to receiving your revised manuscript.

Kind regards,

Ebrahim Shokoohi

Academic Editor

PLOS ONE

Additional Editor Comments (if provided):

Dear Dr Malan

I have checked the manuscript, and despite the improvement of many parts of the manuscript, the Discussion still needs to be improved. The referee commented on the Discussion to be improved based on your finding. Revisions 2 and 3 submitted previously have not significant differences. Therefore, I will suggest you improve the Discussion for finalizing your manuscript.

Kind regards,

Ebrahim Shokoohi

---

## [Author Response · Author response to Decision Letter 2]

19 Jan 2021

The statement "Soil samples were collected from the orchards surveyed plum, apricot and quince, at a depth of 15 cm, from under the tree canopy on Tierhoek farm (-33˚13'19.687''S 19˚38'44.281'') (- 33,711596; 19,790730), in Tierhoek Valley near Robertson in the Western Cape province. A permit for collection of soil samples at the farm was issued by the farm owner and manager B.K.C. Gilson." has been added at the beginning of the Methods section of the manuscript as suggested by Anna Mentsl.

---

## [Editor Report · Decision Letter 3]

1 Feb 2021

PONE-D-20-31254R3

Characterization of Metarhizium majus (Hypocreales: Clavicipitaceae) isolated from the Western Cape province, South Africa

PLOS ONE

Dear Dr. Malan,

Thank you for submitting your manuscript to PLOS ONE. After careful consideration, we feel that it has merit but does not fully meet PLOS ONE’s publication criteria as it currently stands. Therefore, we invite you to submit a revised version of the manuscript that addresses the points raised during the review process.

We look forward to receiving your revised manuscript.

Kind regards,

Ebrahim Shokoohi

Academic Editor

PLOS ONE

Journal Requirements:

Additional Editor Comments (if provided):

Dear Prof Malan

I have checked up the manuscript submitted by you and your colleagues. I am pleased to inform you that your manuscript was accepted for publication in PLOS ONE.

Regards,
---

## [Author Response · Author response to Decision Letter 3]

3 Feb 2021

Ebrahim Shokoohi

Dear Prof Malan

I have checked up the manuscript submitted by you and your colleagues. I am pleased to inform you that your manuscript was accepted for publication in PLOS ONE.

---

## [Editor Report · Decision Letter 4]

5 Feb 2021

Characterization of Metarhizium majus (Hypocreales: Clavicipitaceae) isolated from the Western Cape province, South Africa

PONE-D-20-31254R4

Dear Dr. Malan,

We’re pleased to inform you that your manuscript has been judged scientifically suitable for publication and will be formally accepted for publication once it meets all outstanding technical requirements.

Kind regards,

Ebrahim Shokoohi

Academic Editor

PLOS ONE

Additional Editor Comments (optional):

Dear Prof Malan

I have checked the manuscript. Based on the referee and Editor comments that have been implemented in the submitted manuscript, I am pleased to inform you that your manuscript was accepted for publication in PLOS ONE.

Regards,

E

Reviewers' comments:

no comment

---

## [Editor Report · Acceptance letter]

9 Feb 2021

PONE-D-20-31254R4 

Characterisation of *Metarhizium majus* (Hypocreales: Clavicipitaceae) isolated from the Western Cape Province, South Africa 

Dear Dr. Malan:

I'm pleased to inform you that your manuscript has been deemed suitable for publication in PLOS ONE. Congratulations! Your manuscript is now with our production department. 

Kind regards, 

on behalf of

Dr. Ebrahim Shokoohi 

Academic Editor

PLOS ONE